# Adaptive Layer-Wise Transformations for Post-Training Quantization of Large Language Models

## Abstract

Large language models require significant computational resources for deployment, making quantization essential for practical applications. However, the main obstacle to effective quantization lies in systematic outliers in activations and weights, which cause substantial LLM performance degradation, especially at low-bit settings. While existing transformation-based methods like affine and rotation transformations successfully mitigate outliers, they apply the homogeneous transformation setting, i.e., using the same transformation types across all layers, ignoring the heterogeneous distribution characteristics within LLMs. In this paper, we propose an adaptive transformation selection framework that systematically determines optimal transformations on a per-layer basis. To this end, we first formulate transformation selection as a differentiable optimization problem to achieve the accurate transformation type for each layer. However, searching for optimal layer-wise transformations for every model is computationally expensive. To this end, we establish the connection between weight distribution kurtosis and accurate transformation type. Specifically, we propose an outlier-guided layer selection method using robust $z$-score normalization that achieves comparable performance to differentiable search with significantly reduced overhead. Comprehensive experiments on LLaMA family models demonstrate that our adaptive approach consistently outperforms the widely-used homogeneous transformation settings. For example, our method achieves an improvement of up to 4.58 perplexity points and a 2.11% gain in average six-task zero-shot accuracy under aggressive W3A3K2V2 quantization settings for the LLaMA-3-8B model compared to the current best existing method, FlatQuant, demonstrating the necessity of heterogeneous transformation selection for optimal LLM quantization.

## 1 Introduction

Large language models (LLMs) (Wei et al., 2022; Touvron et al., 2023; Zhang et al., 2022), have gained significant attention due to their remarkable performance in handling complex natural language tasks (Hendrycks et al., 2020), such as language generation, translation, question answering, and text summarization. However, their billion-parameter scale requires significant computational resources for inference and deployment (Frantar et al., 2023; Lin et al., 2023). Quantization, especially Post-Training Quantization (PTQ), has emerged as the dominant compression technique, significantly reducing both memory footprint and computational requirements while preserving model capabilities (Dettmers et al., 2022; Xiao et al., 2023).

The primary obstacle to quantize LLMs effectively is the presence of outliers (An et al., 2025) of activations and weight parameters. For example, Wei et al. (2023) points out that activation values that exceed the mean by 10-100 standard deviations appear consistently at specific channels across different inputs. This causes significant quantization degradation, especially when quantized to extreme low-bit. Several methods have been proposed to address this challenge. In (Ashkboos et al., 2024a; Cui & Wang, 2024; Kim et al., 2023), they propose mixed-precision, which assigns some of the channels to higher precision and less sensitive channels to lower precision to balance accuracy and efficiency. Another approach is to apply transformations to the weights and activations to mitigate the outliers problem. SmoothQuant (Xiao et al., 2023), OmniQuant (Shao et al., 2024)

apply per-channel scaling to balance activation and weight magnitudes. However, these methods cannot effectively quantize the LLMs to 4-bit weights and activations. Affine transformation (Ma et al., 2024; Sun et al., 2025) and rotation transformation (Ashkboos et al., 2024b; Liu et al., 2025; Hu et al., 2025) are proposed to mitigate the outliers problem effectively and successfully quantize the LLMs to 4-bit setting. Rotation transformation-based approaches (Ashkboos et al., 2024b; Liu et al., 2025; Hu et al., 2025) apply orthogonal matrices to spread outliers across dimensions to mitigate the outliers problem. Meanwhile, affine transformation-based approaches (Ma et al., 2024; Sun et al., 2025) apply learnable matrices to flatten the distribution to mitigate the outliers problem. While affine transformations theoretically offer greater flexibility than rotation transformations for handling outliers, the original AffineQuant (Ma et al., 2024) approach has practical limitations. It learns a full transformation matrix that can only be applied to output projection layers for weight-activation quantization, where it merges with preceding linear layers to avoid overhead. Other layers must use per-channel scaling, limiting the method's broader applicability across model architectures. FlatQuant(Sun et al., 2025) uses the Kronecker product to decompose the transformation into two smaller matrices, significantly reducing the number of learnable parameters and achieving the best performance among the existing transformation-based approaches for PTQ for LLMs. Even though these approaches successfully quantize LLMs to 4 bits with slight performance degradation, they apply the same type of transformation across all layers, ignoring the distribution characteristics of each layer within LLMs.

Our key insight is that different layers exhibit fundamentally different statistical properties that determine their optimal transformation. Each layer has its own distribution characteristics; therefore, the optimal transformation type could be different for different layers. Specifically, we propose an approach using differentiable search to find the optimal transformation type for each layer. However, this approach is computationally expensive and not practical for large-scale models. To that end, we propose using outliers as a critical statistical measure for transformation selection. Specifically, we leverage Kurtosis (DeCarlo, 1997) to analyze weight outliers and scrutinize the relationship between kurtosis and the optimal transformation types. We observe that layers with high kurtosis contain concentrated outliers that benefit from rotation's redistribution capability, while layers with low kurtosis have naturally flat distributions where efficient affine transformations suffice. We empirically found that attention layers typically exhibit high outliers, often favor rotation, while certain FFN layers show lower outliers compared to attention layers, often favor affine transformations. Based on this insight, we propose an outlier-guided transformation selection approach that leverages the kurtosis-based insights to efficiently identify the best transformation type for each layer.

In summary, the main contributions of this paper are as follows.

- Unlike existing transformation-based approaches for handling outliers in LLM quantization, which typically apply the same transformation types across all layers, we introduce the novel idea of using layer-specific transformations and demonstrate the superior performance across multiple model architectures and quantization settings.
- We provide the connection between weight outliers and the optimal transformation type and propose an efficient heuristic outlier-guided transformation selection method that achieves comparable performance to differentiable search with significantly reduced computational overhead, making it practical for large-scale models where differentiable search becomes computationally prohibitive.

## 2 RELATED WORKS AND BACKGROUND

### 2.1 RELATED WORKS

**Post-Training Quantization.** Post-training quantization (PTQ) has become the dominant approach for compressing LLMs due to its efficiency and practicality compared to quantization-aware training. These methods apply quantization to pre-trained models using minimal calibration data, avoiding expensive retraining. GPTQ (Frantar et al., 2023) pioneered layer-wise quantization using second-order Hessian information for error compensation, achieving impressive compression rates. AWQ (Lin et al., 2023) advances weight-only quantization by incorporating activation awareness, identifying salient weight channels based on activation magnitudes. Other notable PTQ methods include ZeroQuant (Yao et al., 2022), which proposes fine-grained quantization schemes, and

OWQ (Lee et al., 2024), which improves upon AWQ's activation-aware approach. Recent works like QuIP (Chee et al., 2023) and QuIP# (Tseng et al., 2024) introduce vector quantization with incoherent processing, though at the cost of additional computational overhead.

**Transformation-Based Approaches**  Transformation-based methods that modify model weights and activations are the most effective approaches for mitigating outliers in LLM quantization. SmoothQuant (Xiao et al., 2023) applies per-channel scaling to balance quantization difficulty between activations and weights. Rotation transformations offer several key advantages for outlier mitigation. They preserve the Euclidean norm of weight vectors, maintaining the geometric structure of the original model. Additionally, they redistribute concentrated outliers across multiple dimensions through orthogonal matrix multiplication, effectively flattening the distribution without information loss. QuaRot (Ashkboos et al., 2024b) is the first to apply rotation transformation in post-training quantization (PTQ) for LLMs and successfully quantizes LLMs to 4-bit weights and activations. Specifically, they adopt the Hadamard transform to redistribute outliers across dimensions. SpinQuant (Liu et al., 2025) and OSTQuant (Hu et al., 2025) further improve performance by learning optimal orthogonal matrices to better mitigate the outlier problem. Meanwhile, affine transformations provide greater flexibility in reshaping distributions compared to rotation transformations. AffineQuant (Ma et al., 2024) is the first to propose learnable affine transformations for outlier mitigation. FlatQuant (Sun et al., 2025) further improves this approach by proposing a Kronecker-based affine transformation to reduce the number of learnable parameters and achieve significant improvements, becoming the best existing transformation-based approach.

**Statistical Analysis in Quantization**  Understanding the statistical properties of weight and activation distributions is fundamental to effective quantization design. Prior work has shown that outliers in LLMs exhibit systematic patterns (Wei et al., 2023), consistently emerging in specific channels across different inputs, with their magnitude and frequency strongly correlating with model scale and becoming increasingly pronounced beyond 6.7B parameters. However, existing research focuses primarily on outlier identification rather than understanding how statistical properties should guide transformation selection. Our work addresses this gap by investigating how the statistical characteristics of outliers can be used to guide transformation selection. While kurtosis has been recognized as an indicator of distribution tailedness, the relationship between distribution characteristics and optimal transformation selection remains largely unexplored. We address this limitation by establishing kurtosis as the critical statistical measure for transformation selection.

## 2.2 BACKGROUND

Given a pre-trained LLM with full-precision, a layer of weights $\mathbf{W} \in \mathbb{R}^{m \times n}$ and an input activations $\mathbf{X} \in \mathbb{R}^{b \times m}$, quantization maps these to lower-precision representations. The quantized output of the layer is given by:

$$\hat{\mathbf{Y}} = \mathcal{Q}_a(\mathbf{X}) \cdot \mathcal{Q}_w(\mathbf{W}), \tag{1}$$

where $\mathcal{Q}_a$ and $\mathcal{Q}_w$ denote the quantization functions of the activations and weights, respectively. For a $k$-bit quantization, these functions can be expressed as follows:

$$\mathcal{Q}(\mathbf{Z}) = s \cdot \text{clip}\left(\text{round}\left(\frac{\mathbf{Z}}{s}\right), -2^{k-1}, 2^{k-1} - 1\right), \tag{2}$$

where $s$ is the quantization scaling factor.

Affine (Ma et al., 2024; Sun et al., 2025) and rotation (Ashkboos et al., 2024b; Liu et al., 2025; Hu et al., 2025) are the two important transformation approaches to deal with the outliers for post-training quantization for LLMs. The affine transformation flattens distributions using a learnable matrix:

$$\hat{\mathbf{Y}}_A = \mathcal{Q}_a(\mathbf{X}\mathbf{A})\mathcal{Q}_w(\mathbf{A}^{-1}\mathbf{W}), \tag{3}$$

where $\mathbf{A}$ is a learnable affine transformation matrix.

The rotation transformation uses orthogonal matrices for outlier redistribution:

$$\hat{\mathbf{Y}}_R = \mathcal{Q}_a(\mathbf{X}\mathbf{R})\mathcal{Q}_w(\mathbf{R}^\top\mathbf{W}), \tag{4}$$

where $\mathbf{R} \in \mathbb{R}^{m \times m}$ satisfies $\mathbf{R}^\top\mathbf{R} = \mathbf{I}$.

We demonstrate that different layers exhibit distinct statistical properties that make them more suitable for different transformation types. This motivates our adaptive selection approach.

## 3 PROPOSED METHOD

We present a novel transformation selection framework for post-training quantization (PTQ) of large language models (LLMs). We first present the motivation for adaptive transformation selection, then introduce a layer-wise transformation selection method using differentiable search. While effective, this approach is computationally expensive and impractical for large-scale applications. To address this limitation, we analyze the differentiable search results to identify correlations between weight distribution characteristics and selected transformation types. These insights enable us to develop a more efficient heuristic approach that not only maintains effectiveness but is also practical for large-scale models. Our approach combines both insights from statistical analysis and practical techniques to achieve state-of-the-art quantization performance.

### 3.1 PRELIMINARY ANALYSIS ON THE IMPACT OF THE TRANSFORMATION SELECTION

Table 1: Comparison of the performance of adaptive transformation selection using LLaMA-2-7B for W3A3K3V3 quantization setting. We report the mean and standard deviation of the performance over 20 random sets of the transformation selection as well as the best result from 20 random trials.

| Configuration | WikiText-2 ($\downarrow$) | C4 ($\downarrow$) | Zero-shot Avg ($\uparrow$) |
|---|---|---|---|
| FP16 | 5.47 | 7.26 | 69.79 |
| Fixed Affine Transformation | 7.54 | 9.76 | 59.89 |
| Fixed Rotation Transformation | 7.99 | 10.90 | 58.91 |
| Random Transformation Selection | $7.61 \pm 0.36$ | $9.96 \pm 0.59$ | $59.64 \pm 0.75$ |
| **Best result** | **7.26** | **9.42** | **61.15** |

We conduct a preliminary analysis to study the effect of transformation matrices in post-training quantization (PTQ) for LLMs. We use the released code from FlatQuant (Sun et al., 2025) to quantize LLaMA-2-7B with the Affine transformation. To study the effect of the transformation matrices, we randomly assign 50% of layers of attention and feedforward layers to use the Affine transformation and the remaining 50% to use the Rotation transformation. As shown in Table 1, the best result among 20 runs improves over the setting using all Affine transformations by 0.28 and 0.16 perplexity scores on WikiText2 (Merity et al., 2016) and C4 (Raffel et al., 2020), respectively. Moreover, the performance on average accuracy across six zero-shot tasks significantly improves by 1.26 points. This implies that performance can be significantly improved by carefully selecting the transformation for each layer when performing post-training quantization for LLMs.

### 3.2 DIFFERENTIABLE SEARCH FOR ACCURATE TRANSFORM SELECTION

To preliminarily assess the potential performance of layer-wise adaptive transformation, we formulate the transformation selection as a differentiable search problem. For each layer $l$, the parameters $\boldsymbol{\alpha}^{(l)} = [\alpha_A^{(l)}, \alpha_R^{(l)}]^\top$ control the mixture between affine and rotation transforms.

The quantized output for layer $l$ is:

$$\hat{\mathbf{Y}}^{(l)} = \pi_A^{(l)} \cdot \hat{\mathbf{Y}}_A^{(l)} + \pi_R^{(l)} \cdot \hat{\mathbf{Y}}_R^{(l)}, \tag{5}$$

where $\pi_t^{(l)} = \exp\left(\alpha_t^{(l)}\right) / \sum_{t'} \exp\left(\alpha_{t'}^{(l)}\right)$ are softmax weights ensuring $\pi_A^{(l)} + \pi_R^{(l)} = 1$.

We optimize a loss function that combines reconstruction error and entropy regularization:

$$\mathcal{L} = \sum_l \left[ \mathcal{L}_{\text{recon}}^{(l)} + \lambda_{\text{entropy}} \mathcal{H}(\boldsymbol{\pi}^{(l)}) \right], \tag{6}$$

where $\mathcal{L}_{\text{recon}}^{(l)} = \|\mathbf{Y}^{(l)} - \hat{\mathbf{Y}}^{(l)}\|_F^2$ measures the reconstruction error between the full-precision layer output $\mathbf{Y}^{(l)}$ and the quantized output $\hat{\mathbf{Y}}^{(l)}$, $\mathcal{H}(\boldsymbol{\pi}^{(l)}) = -\sum_t \pi_t^{(l)} \log \pi_t^{(l)}$ is the entropy regularization that encourages the softmax weights to converge toward binary values (zero or one), and $\lambda_{\text{entropy}}$ is a hyper-parameter weighting the contribution of the entropy.

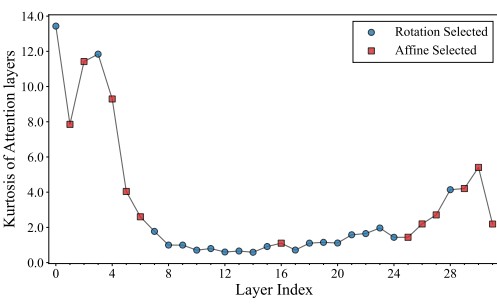
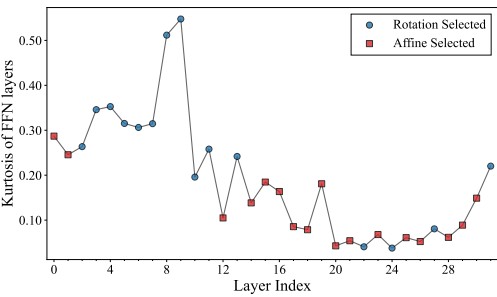

(a) Kurtosis of Attention layers in LLaMA-2-7B     (b) Kurtosis of FFN layers in LLaMA-2-7B

Figure 1: Kurtosis scores across layers for attention and FFN components with corresponding selected transformations using differentiable search. Blue circles indicate layers where rotation transformation was selected, while red squares indicate layers where affine transformation was selected based on differentiable search.

After convergence, we discretize the selection:

$$\mathbf{T}^{(l)} = \begin{cases} \mathbf{A}^{(l)} & \text{if } \arg\max_t \pi_t^{(l)} = A \\ \mathbf{R}^{(l)} & \text{if } \arg\max_t \pi_t^{(l)} = R. \end{cases} \tag{7}$$

While differentiable search can achieve accurate transformation selection, its computational overhead makes it impractical for large-scale models. Therefore, we develop an efficient statistical framework to guide transformation selection based on layer-specific distribution properties.

### 3.3 CORRELATION BETWEEN KURTOSIS AND TRANSFORMATION SELECTION

While differentiable search yields accurate per-layer transform choices, it is computationally expensive for large models and potentially unnecessary. To that end, we propose an outlier-guided layer-wise transformation selection that leverages the *kurtosis* – a statistical measure of a distribution's "tailedness" – as an indicator to efficiently choose the best transformation for each layer.

We compute kurtosis as a metric to characterize weight distribution properties. For layer $l$ with weights $\mathbf{W}^{(l)}$, the excess kurtosis is:

$$\kappa^{(l)} = \frac{\mathbb{E}[(\text{vec}(\mathbf{W}^{(l)}) - \mu^{(l)})^4]}{(\sigma^{(l)})^4} - 3, \tag{8}$$

where $\mu^{(l)}$ and $\sigma^{(l)}$ are the mean and standard deviation of the vectorized weights and $\text{vec}(.)$ is the function flattening a matrix into a vector.

A high value of the kurtosis (i.e., leptokurtic distributions) indicates heavy tails and the presence of outliers, while a low value of the kurtosis (i.e., platykurtic distributions) suggests a more uniform distribution (DeCarlo, 1997). We adopt kurtosis because it is tail-sensitive, directly reflects outliers, and correlates with the transformation choices found via differentiable search. This makes kurtosis a practical diagnostic for anticipating which layers benefit most from rotation or affine transforms.

To quantify the relationship between outliers and transformation selection, we compute kurtosis scores for each layer type. For attention layers, we calculate the sum of the kurtosis score of Query (Q), Key (K), and Value (V) layers, while for feedforward (FFN) layers, we compute the kurtosis score of the Gate/Up projection layer using Eq. 8. These kurtosis scores capture the concentration of extreme values in each layer type and provide a quantitative basis for understanding the correlation between statistical properties and transformation selection.

Specifically, Figure 1 highlights a correlation between the kurtosis scores and the selected transformation. For the attention layers, layers with higher kurtosis scores are more likely to be selected for the affine transformation, while most layers with kurtosis scores less than 2.0 are selected for the rotation transformation. Conversely, for the FFN layers, layers with higher kurtosis scores are

more likely to be selected for the rotation transformation, while most layers with kurtosis scores less than 0.2 are selected for the affine transformation. Additionally, for each type of layer, the rotation or affine transformations are selected on both high and low kurtosis scores, indicating that kurtosis provides guidance rather than strict thresholds for transformation selection.

To bridge the gap between statistical understanding and practical implementation, we propose an outlier-guided layer-wise transformation selection that leverages the kurtosis metric as an indicator to efficiently identify the best transformation for each layer.

### 3.4 OUTLIER-GUIDED TRANSFORMATION SELECTION

Given the model comprises $n$ sequential attention (or feedforwad) layers indexed $i \in \{0, \ldots, n-1\}$, we treat kurtosis value as an outlier indicator and set $o_i = |\kappa^{(i)}|$, which is the absolute value of the kurtosis score of the layer as the layer's outlier score. Our goal is to leverage $\{o_i\}$ to select $L$ ($1 \le L \le n$) layer indices as the rotation transformation selection and the remaining $n - L$ layer indices as the affine transformation selection.

**Robust normalization.** To achieve scale- and skew-robustness, we convert the absolute kurtosis scores $\{o_i\}$ to the robust $z$-scores (Iglewicz & Hoaglin, 1993) via the median and the median absolute deviation (MAD):

$$\tilde{o}_i = \frac{o_i - \text{median}(o)}{1.4826 \, \text{MAD}(o) + \varepsilon}, \qquad \text{MAD}(o) = \text{median}|o - \text{median}(o)|, \tag{9}$$

with a small value $\varepsilon > 0$ (e.g., $10^{-12}$) for numerical stability. The factor $1.4826$ scales MAD to be comparable to a standard deviation under normality.

Empirically, rotation transformations are concentrated in both high and low kurtosis scores, with a prevalence in higher kurtosis scores for FFN layers and a prevalence in lower kurtosis scores for attention layers. We therefore allocate a fraction $\beta$ of the $L$ layers as the rotation transformation to the higher part of kurtosis scores:

$$K_{\text{high}} = \lfloor \beta L \rceil, \qquad K_{\text{low}} = L - K_{\text{high}}, \tag{10}$$

where $\lfloor . \rceil$ denotes the rounding-to-nearest function.

Optionally, $\beta$ can be set by the positive-vs-absolute $z$-mass: For rotation transformation selection at attention layers, we set $\beta_{\text{attn}}$ as:

$$\beta_{\text{attn}} = \text{clip}\left(\frac{\sum_{i:\tilde{o}_i>0} \tilde{o}_i}{\sum_i |\tilde{o}_i|}, 0.1, 0.3\right). \tag{11}$$

For rotation transformation selection at FFN layers, we set $\beta_{\text{ffn}}$ as:

$$\beta_{\text{ffn}} = \text{clip}\left(\frac{\sum_{i:\tilde{o}_i>0} \tilde{o}_i}{\sum_i |\tilde{o}_i|}, 0.7, 0.9\right). \tag{12}$$

We define upper and lower thresholds using order statistics of the outlier scores $\{\tilde{o}_i\}_{i=0}^{n-1}$. For the upper tail, we select the $(n - K_{\text{high}})$-th largest value as the threshold:

$$\tau_{\text{high}} = \begin{cases} (n - K_{\text{high}})\text{-th largest value in } \{\tilde{o}_i\}_{i=0}^{n-1} & \text{if } K_{\text{high}} > 0 \\ +\infty & \text{if } K_{\text{high}} = 0 \end{cases} \tag{13}$$

Similarly, for the lower tail, we select the $(K_{\text{low}})$-th smallest value:

$$\tau_{\text{low}} = \begin{cases} (K_{\text{low}})\text{-th smallest value in } \{\tilde{o}_i\}_{i=0}^{n-1} & \text{if } K_{\text{low}} > 0 \\ -\infty & \text{if } K_{\text{low}} = 0 \end{cases} \tag{14}$$

The candidate index set for the selection of rotation transformation is the union of the two tails:

$$\mathcal{C} = \{ i : \tilde{o}_i \ge \tau_{\text{high}} \} \cup \{ i : \tilde{o}_i \le \tau_{\text{low}} \}. \tag{15}$$

The heuristic seamlessly integrates with existing quantization approaches, requiring only statistical classification to determine transform types and then applicable for large-scale models compared to the differentiable search.

## 4 EXPERIMENTS

### 4.1 EXPERIMENTAL SETUP

**Models and Datasets.** We evaluate on the LLaMA family: LLaMA-2 (7B, 13B, 70B) (Touvron et al., 2023) and LLaMA-3 (8B, 70B), covering diverse model scales. Following previous works (Shao et al., 2024; Ashkboos et al., 2024b), we report perplexity (PPL) on WikiText2 (Merity et al., 2016) and C4 (Raffel et al., 2020) test sets for language modeling evaluation. For downstream task evaluation, we assess model performance on six zero-shot tasks using the `lm-evaluation-harness` framework, including ARC-Easy and ARC-Challenge (Clark et al., 2018), HellaSwag (Zellers et al., 2019), LAMBADA (Paperno et al., 2016), PIQA (Bisk et al., 2020), and WinoGrande (Sakaguchi et al., 2021). We report the average accuracy across six zero-shot tasks as the primary evaluation metric for downstream performance. For calibration, we randomly sample 128 sequences, each containing 2048 tokens, from the WikiText-2 dataset (Merity et al., 2016).

**Baselines.** We compare against state-of-the-art PTQ methods representing different transformation approaches: SmoothQuant (Xiao et al., 2023), affine transformation methods FlatQuant (Sun et al., 2025), and rotation-based transformation methods QuaRot (Ashkboos et al., 2024b), SpinQuant (Liu et al., 2025), and OSTQuant (Hu et al., 2025). These baselines represent the spectrum of current PTQ approaches, from traditional weight quantization to advanced transformation-based methods.

**Implementation Details.** Our method is implemented in PyTorch with HuggingFace Transformers. For the rotation transformation, we initialize the rotation matrix with random orthogonal matrices and adopt RiemannAdam (Becigneul & Ganea, 2019) as the optimizer for the learnable rotation matrix. For the learnable affine transformation, we use the same implementation as FlatQuant (Sun et al., 2025). We use the AdamW optimizer with a learning rate of $5 \times 10^{-3}$. For differentiable search, we set the entropy regularization to $\lambda_{\text{entropy}} = 0.01$. We employ symmetric per-channel weight quantization and per-token activation quantization with GPTQ weight quantizers following existing approaches (Sun et al., 2025; Hu et al., 2025). Following (Sun et al., 2025), we also employ the combination of scaling transformation with the selected transformation and adopt learnable clipping thresholds (Shao et al., 2024) for weights and activations to eliminate outliers. For all experiments, we empirically set $\beta_{\text{attn}} = 0.1$ and $\beta_{\text{ffn}} = 0.9$ for the outlier-guided transformation selection method. The parameter $L$ in Eq. 10 is set to $0.7 \times n$ for attention layers and set to $0.5 \times n$ for feedforward layers, where $n$ is the number of model attention (or feedforward) layers. We conduct an ablation study on these hyperparameters in Appendix A.1. The notation *W4A4K4V4* denotes quantization with 4-bit weights and 4-bit activations, where *K4* and *V4* indicate 4-bit quantization for the key and value projection layers in attention modules, respectively. For the positioning of the adaptive transformation, we selectively apply the proposed affine transformation to QKV layers in attention modules and up-gate layers in FFN modules. For all other linear layers, we follow the FlatQuant approach (Sun et al., 2025). This selective design choice preserves runtime speedup while achieving significant performance improvements over FlatQuant.

### 4.2 EXPERIMENTAL RESULTS

**Evaluation on generation datasets with perplexity.** In this section, we compare our proposed adaptive transformation selection method against state-of-the-art PTQ approaches, including QuaRot (Ashkboos et al., 2024b), SpinQuant (Liu et al., 2025), FlatQuant (Sun et al., 2025), and OSTQuant (Hu et al., 2025). The results of these methods are reproduced using their official implementations and are cited from FlatQuant (Sun et al., 2025). Table 2 presents the comparative results of our proposed method and other state-of-the-art approaches when evaluating on WikiText-2 and C4 datasets. Our results are obtained using the proposed efficient heuristic outlier-guided transformation selection approach. We conduct experiments across multiple quantization configurations: W4A4K4V4, W3A3K3V3, W4A4K2V2, and W3A3K2V2. In general, our proposed method outperforms the state-of-the-art methods across various model architectures and quantization settings. Compared to the best existing method, FlatQuant (Sun et al., 2025), our proposed method consistently outperforms FlatQuant in all bit-width configurations. The improvement is clearer in extreme quantization settings, with improvements over FlatQuant of 0.55 and 0.80 points for LLaMA-3-8B in the W3A3K3V3 setting on WikiText-2 and C4 datasets respectively. Under the extreme

Table 2: Perplexity scores (↓) on WikiText-2 and C4 datasets for various quantization settings on LLaMA models.

| Setting | Method | WikiText-2 (↓) | | | C4 (↓) | | |
|---|---|---|---|---|---|---|---|
| | | 2-7B | 2-13B | 3-8B | 2-7B | 2-13B | 3-8B |
| | FP16 | 5.47 | 4.88 | 6.14 | 7.26 | 6.73 | 9.45 |
| W4A4KV4 | QuaRot | 6.10 | 5.40 | 8.16 | 8.32 | 7.54 | 13.38 |
| | SpinQuant | 5.96 | 5.24 | 7.39 | 8.28 | 7.48 | 12.19 |
| | OSTQuant | 5.91 | 5.25 | 7.29 | – | – | – |
| | FlatQuant | 5.78 | 5.11 | 6.90 | 7.86 | 7.11 | 11.21 |
| | **Ours** | **5.61** | **4.97** | **6.89** | **7.58** | **6.74** | **10.08** |
| W3A3K3V3 | OSTQuant | 8.32 | 6.86 | 13.59 | 11.42 | 9.45 | 21.06 |
| | FlatQuant | 7.54 | 6.14 | 10.62 | 9.76 | 8.15 | 16.20 |
| | **Ours** | **7.22** | **6.02** | **10.07** | **9.43** | **8.08** | **15.40** |
| W4A4K2V2 | OSTQuant | 8.32 | 6.65 | 12.88 | 11.14 | 9.03 | 19.84 |
| | FlatQuant | 7.51 | 5.98 | 8.73 | 9.87 | 8.11 | 13.86 |
| | **Ours** | **6.98** | **5.79** | **8.25** | **9.04** | **7.66** | **12.59** |
| W3A3K2V2 | OSTQuant | 12.28 | 8.97 | 26.48 | 17.36 | 12.65 | 43.62 |
| | FlatQuant | 11.51 | 7.86 | 16.38 | 15.89 | 10.91 | 27.29 |
| | **Ours** | **9.83** | **7.15** | **13.74** | **13.35** | **9.76** | **22.71** |

Table 3: Zero-shot QA task results of 4-bit and 3-bit weight & activation quantized LLaMA models.

| Model | Method | ARC-C (↑) | ARC-E (↑) | HellaSwag (↑) | LAMBADA (↑) | PIQA (↑) | WinoGrande (↑) | Avg (↑) |
|---|---|---|---|---|---|---|---|---|
| | | | | *W4A4KV4 Settings* | | | | |
| LLaMA-2-7B | FP16 | 46.16 | 74.54 | 75.98 | 73.92 | 79.05 | 69.06 | 69.79 |
| | QuaRot | 42.32 | 68.35 | 72.53 | 65.40 | 76.33 | 65.11 | 65.01 |
| | SpinQuant | 41.72 | 69.28 | 72.90 | 71.28 | 76.17 | 66.06 | 66.23 |
| | FlatQuant | 43.00 | 71.21 | 73.31 | **72.06** | 77.53 | 67.72 | 67.47 |
| | Ours | **43.94** | **72.31** | **73.66** | 71.38 | **77.64** | **68.35** | **67.88** |
| LLaMA-2-13B | FP16 | 49.15 | 77.44 | 79.39 | 76.73 | 80.47 | 72.14 | 72.55 |
| | QuaRot | 45.48 | 73.27 | 76.03 | 69.01 | 79.05 | 70.64 | 68.91 |
| | SpinQuant | 49.15 | 77.19 | 76.86 | 73.86 | 78.67 | 69.85 | 70.93 |
| | FlatQuant | 48.38 | **76.94** | 77.88 | **76.40** | 79.65 | 70.56 | 71.64 |
| | Ours | **49.91** | 75.93 | **77.92** | 76.03 | **79.71** | 70.64 | **71.69** |
| LLaMA-3-8B | FP16 | 53.50 | 77.57 | 79.12 | 75.51 | 80.74 | 72.93 | 73.23 |
| | QuaRot | 45.73 | 70.83 | 72.97 | 62.70 | 75.35 | 67.17 | 65.79 |
| | SpinQuant | 47.27 | 74.20 | 74.55 | 70.29 | 77.37 | 68.51 | 68.70 |
| | FlatQuant | 50.51 | 75.88 | 76.49 | **73.20** | **79.00** | **72.93** | 71.33 |
| | Ours | **51.45** | **78.16** | **76.50** | 72.50 | 78.89 | 72.38 | **71.65** |
| | | | | *W3A3K3V3 Settings* | | | | |
| LLaMA-2-7B | OSTQuant | 30.72 | 55.09 | 60.21 | 57.05 | 68.44 | 57.77 | 54.88 |
| | FlatQuant | 36.35 | 60.82 | 66.20 | 60.59 | 73.29 | 59.43 | 59.45 |
| | Ours | **36.52** | **63.17** | **66.42** | **62.91** | **74.43** | **63.61** | **61.18** |
| LLaMA-2-13B | OSTQuant | 37.37 | 61.41 | 66.44 | 62.12 | 71.98 | 58.01 | 59.56 |
| | FlatQuant | 40.96 | 70.41 | 71.92 | 71.41 | **77.20** | **68.90** | 66.80 |
| | Ours | **42.83** | **71.97** | **72.33** | **71.71** | 76.77 | 66.14 | **66.94** |
| LLaMA-3-8B | OSTQuant | 31.83 | 52.15 | 56.72 | 45.64 | 66.49 | 55.09 | 51.32 |
| | FlatQuant | 37.46 | 63.09 | 64.86 | **55.54** | 71.17 | 61.40 | 59.01 |
| | Ours | **38.05** | **65.19** | **65.71** | 52.73 | **73.29** | **62.98** | **59.66** |
| | | | | *W3A3K2V2 Settings* | | | | |
| LLaMA-2-7B | OSTQuant | 26.54 | 46.46 | 48.11 | 34.45 | 64.58 | 52.64 | 45.46 |
| | FlatQuant | 30.89 | 50.63 | 56.79 | 32.95 | 67.08 | 52.72 | 48.51 |
| | Ours | **32.85** | **57.37** | **59.89** | **47.45** | **71.98** | **54.46** | **54.00** |
| LLaMA-2-13B | OSTQuant | 31.66 | 52.69 | 58.43 | 45.41 | 67.36 | 53.28 | 51.47 |
| | FlatQuant | 36.18 | 64.18 | 65.80 | 52.57 | 73.72 | 59.12 | 58.60 |
| | Ours | **37.71** | **65.82** | **66.65** | **58.49** | **73.23** | **60.14** | **60.34** |
| LLaMA-3-8B | OSTQuant | 24.74 | 38.89 | 40.18 | 18.07 | 56.75 | 49.01 | 37.94 |
| | FlatQuant | 33.02 | 56.23 | 55.48 | 25.79 | 69.31 | 56.75 | 49.43 |
| | Ours | **34.64** | **56.90** | **57.86** | **32.25** | **69.75** | **57.85** | **51.54** |

low-bitwidth W3A3K2V2 quantization setting, our proposed method demonstrates substantial gains over FlatQuant with improvements of 1.68 and 2.54 perplexity points for LLaMA-2-7B, and 2.64 and 4.58 points for LLaMA-3-8B on WikiText-2 and C4 datasets respectively, which confirms the effectiveness of our adaptive transformation selection approach.

**Evaluation on downstream tasks.** Table 3 presents the comparative results of our proposed method and other state-of-the-art approaches across different quantization settings. The table eval-

uates performance on six tasks: ARC-Challenge, ARC-Easy, HellaSwag, LAMBADA, PIQA, and WinoGrande, with results averaged across all tasks. Under the W3A3K3V3 setting, our method demonstrates significant improvements over existing approaches. For LLaMA-2-7B, our approach achieves an average accuracy of 61.18%, representing a substantial 1.73% improvement over FlatQuant and a 6.30% improvement over OSTQuant. The improvement is clearer in extreme W3A3K2V2 quantization settings, with improvements of 5.49% and 8.54% for LLaMA-2-7B in the W3A3K2V2 setting compared to FlatQuant and OSTQuant, respectively. For LLaMA-2-13B, our method achieves 60.34%, outperforming FlatQuant by 1.74% and OSTQuant by 8.87%. These results highlight the effectiveness of our adaptive transformation selection method in preserving model performance on downstream tasks, especially under extreme quantization scenarios.

## 4.3 ABLATION STUDIES

**The comparison between the proposed heuristic and the differentiable search selection.** Table 4 shows that the proposed outlier-guided transformation selection achieves 87.5% and 85.0% agreement with learned selection for LLaMA-2-7B and LLaMA-2-13B, respectively. Importantly, the proposed heuristic significantly reduces the training time by approximately $3\times$ compared to the learned selection while maintaining competitive performance.

Table 4: Comparison of the proposed heuristic and differentiable search selection on LLaMA-2-7B and LLaMA-2-13B with W3A3K3V3 settings.

| Model | WikiText-2 PPL | | C4 PPL | | Zero-shot Avg | | Selection Agreement | | Training Time (h) | |
|---|---|---|---|---|---|---|---|---|---|---|
| | Learned | Heuristic | Learned | Heuristic | Learned | Heuristic | Layers | Percentage | Learned | Heuristic |
| LLaMA-2-7B | 7.18 | 7.22 | 9.35 | 9.43 | 61.45 | 61.18 | 28/32 | 87.5% | 11 hours | 4 hours |
| LLaMA-2-13B | 5.99 | 6.02 | 8.03 | 8.08 | 67.04 | 66.94 | 34/40 | 85.0% | 22 hours | 7.5 hours |

Table 5: Prefill and decoding speedup comparison on LLaMA2-7B

(a) Prefill speedup compared to FP16 for different input sequence lengths at batch size 1

| Prefill Length | INT4 | QuaRot | FlatQuant | Ours |
|---|---|---|---|---|
| 2048 | $1.36\times$ | $1.19\times$ | $1.37\times$ | $1.32\times$ |
| 4096 | $1.44\times$ | $1.34\times$ | $1.48\times$ | $1.46\times$ |
| 8192 | $1.94\times$ | $1.79\times$ | $1.92\times$ | $1.89\times$ |

(b) Decoding speedup compared to FP16 for different KV cache lengths at batch size 64

| KV Cache Length | INT4 | QuaRot | FlatQuant | Ours |
|---|---|---|---|---|
| 256 | $1.089\times$ | $1.036\times$ | $1.058\times$ | $1.047\times$ |
| 512 | $1.121\times$ | $1.071\times$ | $1.090\times$ | $1.083\times$ |
| 1024 | $1.140\times$ | $1.087\times$ | $1.115\times$ | $1.090\times$ |
| 2048 | $1.167\times$ | $1.106\times$ | $1.141\times$ | $1.117\times$ |

**Speedup across sequence lengths.** Tables 5a and 5b present detailed prefill and decoding speedup results for LLaMA-2-7B under different prefill sequence lengths and KV cache lengths. All experiments were conducted on A100 GPUs. As shown, for prefill, our method achieves up to $1.89\times$ speedup at sequence length 8192, nearly matching FlatQuant's $1.92\times$ performance. For decoding, we achieve up to $1.117\times$ speedup at KV cache length 2048, surpassing QuaRot. These results demonstrate that our adaptive transformation selection approach effectively balances quantization performance and speedup efficiency across different prefill sequence lengths and KV cache lengths.

## 5 CONCLUSION

In this paper, we introduce a novel adaptive framework that selects the optimal transformation on a layer-wise basis. Specifically, we present a differentiable search that formulates the problem as an optimization task to automatically find the best transformation for each layer, and then propose an efficient heuristic method. The heuristic leverages the connection between weight distribution characteristics and the optimal transformation type, achieving performance comparable to the differentiable search with significantly less computational overhead. This makes our approach practical for even the largest models. Our comprehensive experiments on the LLaMA family of models demonstrate the superiority of our adaptive approach. Compared to state-of-the-art methods like FlatQuant, our technique achieves a notable improvement, reducing perplexity by up to 4.58 points on the LLaMA-3-8B model under an aggressive W3A3K2V2 quantization setting. These results validate the importance of layer-specific transformations for effective LLM quantization and offer a scalable solution for practical deployment.

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

**The statement on the use of large language models.** Large Language Models (LLMs) were used solely for grammar correction and language polishing of this manuscript. All research ideas, experimental design and data analysis were conducted entirely by the authors, and the use of LLMs does not impact the reproducibility or validity of our findings.

# A  APPENDIX

## A.1  ABLATION STUDIES OF HYPER-PARAMETERS $\beta_{\text{ATTN}}$, $\beta_{\text{FFN}}$, AND $L$

We conduct ablation studies to analyze the impact of the hyper-parameters $\beta_{\text{attn}}$ and $\beta_{\text{ffn}}$ in Eq. 11 and Eq. 12, respectively, in our adaptive layer-wise transformation scheme for post-training quantization of large language models. These parameters control the percentage of layers that select rotation transformations in the upper tail of z-scores of the attention and feed-forward layers.

Table A.1: Ablation studies for the hyper-parameter $\beta_{\text{attn}}$ of the in Eq. 11. Results are on the LLaMA-2-7B model using W4A4K2V2 quantization.

| $\beta_{\text{attn}}$ | 0.1 | 0.3 | 0.4 | 0.5 | 0.7 | 0.9 |
|---|---|---|---|---|---|---|
| WikiText2 PPL ($\downarrow$) | **7.05** | 7.07 | 7.09 | 7.11 | 7.13 | 7.22 |
| C4 PPL ($\downarrow$) | **9.12** | 9.12 | 9.16 | 9.12 | 9.22 | 9.37 |
| six zero-shot avg ($\uparrow$) | **61.40** | 61.11 | 61.20 | 60.09 | 60.70 | 61.05 |

**Ablation studies for the hyper-parameter $\beta_{\text{attn}}$ in Eq. 11.** For $\beta_{\text{attn}}$, we vary the values from 0.1 to 0.9 and evaluate the model performance on WikiText-2 and C4 datasets using perplexity (PPL) scores, as well as the average zero-shot performance across six downstream tasks. As shown in Table A.2, smaller values of $\beta_{\text{attn}}$ (0.1-0.3) generally yield better performance, with $\beta_{\text{attn}} = 0.1$ achieving the best results across all metrics. This suggests that in attention layers, most selected rotation transformations are from the lower tail of the z-scores distribution.

Table A.2: Ablation studies for the hyper-parameter $\beta_{\text{ffn}}$ of the in Eq. 12. Results are on the LLaMA-2-7B model using W4A4K2V2 quantization.

| $\beta_{\text{ffn}}$ | 0.1 | 0.3 | 0.5 | 0.7 | 0.8 | 0.9 |
|---|---|---|---|---|---|---|
| WikiText2 PPL ($\downarrow$) | 7.30 | 7.26 | 7.25 | 7.21 | 7.23 | **7.08** |
| C4 PPL ($\downarrow$) | 9.54 | 9.46 | 9.41 | 9.41 | 9.42 | **9.37** |
| six zero-shot avg ($\uparrow$) | 60.83 | 61.07 | 61.16 | 61.19 | 61.26 | **61.30** |

**Ablation studies for the hyper-parameter $\beta_{\text{ffn}}$ in Eq.12.** For $\beta_{\text{ffn}}$, we test values ranging from 0.1 to 0.9. As shown in Table A.1, larger values of $\beta_{\text{ffn}}$ (0.7-0.9) generally yield better performance. The results indicate that $\beta_{\text{ffn}} = 0.9$ provides the optimal balance, achieving the lowest perplexity scores and highest zero-shot performance. This finding suggests that the feed-forward layers benefit from selecting rotation transformations predominantly from the upper tail of the z-scores distribution.

**Ablation studies for the hyper-parameter $L$ in Eq. 10.** For the number of layers selecting the rotation transformation parameters $L$, we vary the value of $L$ from $0 \times n$ to $1.0 \times n$, where $n$ is the total number of layers. For attention layers, as shown in Table A.3, the best performance is achieved when $L$ is set to $0.7 \times n$, indicating that selecting rotation transformations for 70% of the total number of attention layers yields best performance. Meanwhile, selecting all the attention layers using rotation transformation could significantly degrade performance. For the feedforward layers, as shown in Table A.4, the best performance is achieved when $L$ is set to $0.5 \times n$, indicating that selecting rotation transformations for 50% of the total number of feedforward layers yields best performance. Similar to attention layers, selecting all the feedforward layers using rotation transformation could also degrade performance.

Table A.3: Ablation studies for the number of selected rotation transformations $L$ for attention layers of the in Eq. 10. Results are on the LLaMA-2-7B model using W3A3K3V3 quantization.

| $L$ | $0 \times n$ | $0.5 \times n$ | $0.7 \times n$ | $1.0 \times n$ |
|---|---|---|---|---|
| WikiText2 PPL (↓) | 7.54 | 7.35 | **7.29** | 8.71 |
| C4 PPL (↓) | 9.76 | 9.66 | **9.62** | 12.01 |
| six zero-shot avg (↑) | 59.89 | 60.09 | **60.37** | 57.67 |

Table A.4: Ablation studies for the number of selected rotation transformations $L$ for feedforward (FFN) layers of the in Eq. 10. Results are on the LLaMA-2-7B model using W3A3K3V3 quantization.

| $L$ | $0 \times n$ | $0.3 \times n$ | $0.5 \times n$ | $0.7 \times n$ | $1.0 \times n$ |
|---|---|---|---|---|---|
| WikiText2 PPL (↓) | 7.54 | 7.45 | **7.41** | 7.49 | 7.51 |
| C4 PPL (↓) | 9.76 | 9.67 | **9.51** | 9.78 | 9.80 |
| six zero-shot avg (↑) | 59.89 | 59.96 | **60.26** | 59.87 | 59.82 |

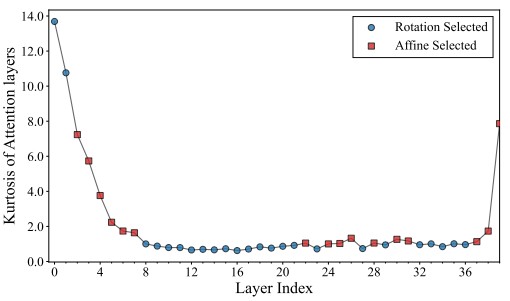
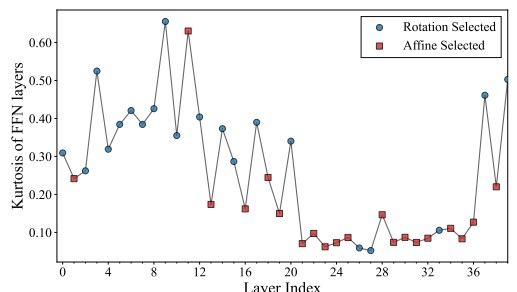

(a) Kurtosis of Attention layers in LLaMA-2-13B    (b) Kurtosis of FFN layers in LLaMA-2-13B

Figure A.1: Kurtosis scores across layers for attention and FFN components with corresponding selected transformations using differentiable search. Blue circles indicate layers where rotation transformation was selected, while red squares indicate layers where affine transformation was selected based on differentiable search.

## A.2 CORRELATION BETWEEN OUTLIERS AND TRANSFORMATION SELECTION USING DIFFERENTIABLE SEARCH

We provide additional analysis on the correlation between outlier statistics, specifically kurtosis scores, and the selected transformations using differentiable search in LLaMA-2-13B model. Specifically, Figure A.1 highlights a correlation between the kurtosis scores and the selected transformation. For the attention layers, layers with higher kurtosis scores are more likely to be selected for the affine transformation, while most layers with kurtosis scores less than $1.6$ are selected for the rotation transformation. Conversely, for the FFN layers, layers with higher kurtosis scores are more likely to be selected for the rotation transformation, while most layers with kurtosis scores less than $0.25$ are selected for the affine transformation. Additionally, for each type of layer, the rotation or affine transformations are selected on both high and low kurtosis scores, indicating that kurtosis provides guidance rather than strict thresholds for transformation selection.

