# OpenReview forum: "Adaptive Layer-Wise Transformations for Post-Training Quantization of Large Language Models"
_ICLR.cc/2026/Conference — ICLR 2026 Conference Withdrawn Submission_

### Official Review · Reviewer_kqyS · 2025-10-26

**Soundness:** 2
**Presentation:** 1
**Contribution:** 1
**Rating:** 2
**Confidence:** 4

**Summary:**

The paper proposes a trick to distinguish layers into Affine/Rotation types, then applies different quantization (FlatQuant/orthogonal rotation) to each layer. Firstly, empirical study indicates that a mixed type of quantization methods can yield better results. Then, a high correlation between a learnable choice factor and selecting via a heuristic metric shows that the metric, i.e., kurtosis, is a more efficient way to select the type of layers. Empirical studies on the LLaMA series further verify the proposed method.

**Strengths:**

1. The proposed heuristic metric is verified thoroughly by various empirical studies.
2. The proposed method is well assessed via experiments.
3. The paper is overall well-structured, making it easy to follow.

**Weaknesses:**

1. Lack of illustrations about how to implement the mixed method. As far as I know, FlatQuant and orthogonal transformation (if keeping computational invariance) are global across layers, how do we combine both of them without conflicts?
2. Lack of implementation details regarding the hardware part. There is no description of how to obtain the statistics in Table 5. In addition, why is the QuaRot method slower than the proposed method, since it is a mixed one that may involve more operations and may break computational invariance?
3. The bold notation in Table 3 is problematic. In the W3A3K2V2 setting, for LLaMA-2-13B, FlatQuant has the best performance on the PIQA dataset (73.72). In addition, in the W4A4KV4 setting, for LLaMA-2-13B, QuaRot yields a result as good as the last line on the Wino dataset (70.64).
4. There is no further analysis of why the proposed heuristic works, and why it behaves differently for FFN/Attention layers. It seems the paper only proposes a heuristic metric without significant contribution beyond empirical observations.

**Questions:**

See weaknesses above.

---

### Official Review · Reviewer_gaPq · 2025-10-31

**Soundness:** 2
**Presentation:** 3
**Contribution:** 3
**Rating:** 4
**Confidence:** 4

**Summary:**

This paper proposes an adaptive selection framework to choose different quantization methods for different layers of LLMs. It models the selection process as an optimization problem and uses z-score normalization to select various methods. The effectiveness of this framework is validated through experiments on the Llama model.

**Strengths:**

1. Good Compatibility: It can be easily integrated with existing methods.
2. Rigorous Experiments: The paper provides detailed parameter analysis and correlation analysis, among others.
3. The paper is well-written and easy to follow.

**Weaknesses:**

1. Limited Innovation: The approach seems to be a simple integration of existing quantization methods through manually defined heuristic rules. The overall potential for improvement is limited.
2. Limited Applicability: The analysis and experimental conclusions are focused on the Llama model family. It is unclear whether the framework is effective on other model families, such as Qwen. Would the heuristic strategy still be effective?
3. Limited Improvement: In the W3A3K3V3 and W4A4K4V4 settings, the improvements of the method are modest. Compared to strong existing baselines, the average improvement is less than 0.5, and the additional computational cost of calculating heuristic indicators results in relatively limited performance gains.
4. Parameter Sensitivity: The parameters \beta_{attn}, \beta_{ffn}, and the values in equations (11) and (12) depend on experimental experience, which could make tuning difficult, especially when transferring to other model families like Qwen.

**Questions:**

Please refer to the weaknesses

---

### Official Review · Reviewer_69rX · 2025-11-01

**Soundness:** 2
**Presentation:** 2
**Contribution:** 2
**Rating:** 2
**Confidence:** 4

**Summary:**

This paper presents a post-training quantization technique that addresses the challenge of outlier values in LLMs. The core idea is to move beyond homogeneous transformation strategies, where all layers use the same transformation method, such as affine or rotation, and instead adopt an adaptive strategy that selects the optimal transformation (affine or rotation) on a per-layer basis. To achieve this, the authors propose a differential search-based algorithm that determines the best transformation strategy for each layer.

**Strengths:**

The empirical results demonstrate significant performance improvements compared to previous methods.

**Weaknesses:**

**Limited Novelty:**

  The proposed method closely resembles a combination of *FlatQuant* and rotation based method, which reduces the perceived novelty of the contribution.

**Clarity and Organization:**

   The paper is difficult to follow, and the connections between the different contributions are not clearly articulated. For example, what they mean by rotation based transformation is it Hadamard Transform or it's also a learnable transform?

 **Limited Experimental Scope:**

   The experiments are conducted solely on the LLaMA model, and all kurtosis-to-transformation mappings are derived from LLaMA. It would strengthen the paper if the authors provided empirical results on other model families such as Qwen or Phi to validate the generality of their approach.

**Lack of Implementation Details and Efficiency Discussion:**

Although the paper reports speed-up results, it does not clarify the implementation scope. Using different transformations (both rotation and affine) across layers seems counterintuitive for efficiency. Previous works achieved speed-ups partly due to the ability to fuse rotations, but applying heterogeneous transformations per layer may require online computation, potentially increasing overhead. The paper lacks details on how this issue is handled, could the authors provide further explanation or clarification?



 **Missing Related Work:**

 Relevant works such as *DartQuant*, "KurTail" are not discussed. Including these in the related work and experimental comparison would provide a more complete evaluation.


- DartQuant: Efficient Rotational Distribution Calibration for LLM Quantization, NeurIPS 2025

- KurTail : Kurtosis-based LLM Quantization, SLLM Workshop ICLR 2025

**Questions:**

Check Weakness

---

### Note · Authors · 2025-11-21

I have read and agree with the venue's withdrawal policy on behalf of myself and my co-authors.